# Simulation Study on Dynamic Thermal Performance of a New Ventilated Roof with Form-Stable PCM in Southern China

**Jinghua Yu, Kangxin Leng, Feifei Wang \*, Hong Ye and Yongqiang Luo**

School of Environmental Science and Engineering, Huazhong University of Science and Technology, Wuhan 430074, China; yujinghua323@126.com (J.Y.); lengkx@hust.edu.cn (K.L.); yh2016subject@163.com (H.Y.); luoyongqiang@hust.edu.cn (Y.L.)

**\*** Correspondence: ffwang@hust.edu.cn; Tel.: +86-132-9416-0016

**Abstract:** Latent heat storage in phase change material (PCM) is an efficient technology that can be applied in building envelopes. Installing PCM in building roof has been effective in altering space cooling loads. However, the heat absorbed by the PCM during the daytime will be released at night; the cooling load is shifted to the night. So, this study proposed a new ventilated roof with form-stable PCM (VRFP). The night cool air is used for ventilation during summer to remove the solidification heat of PCM and to store the cooling energy in the roof. Form-stable PCM is placed in the upper layer and ventilation duct is placed in the middle layer. The inner surface temperature of this roof is reduced sharply compared with the conventional PCM roof. The thermal performance of this PCM roof with night ventilation in Wuhan, a city in southern China, was studied by through Computation Fluid Dynamics (CFD) simulation. A three-dimensional dynamic numerical model of this roof was built. The effects of melting temperature range, thickness of Form-stable PCM layer and ventilation strategy on the thermal performance were analyzed. Results show that, in Wuhan city, the optimal melting temperature range is 35~38 °C, the appropriate thickness of PCM layer is 30~40 mm and the optimal ventilation speed is 2.4~2.5 m/s. This structure can effectively prevent the stored heat of PCM transferring from the exterior to the interior during the summer and reduce cooling energy consumption.

**Keywords:** phase change material; ventilation; thermal performance; melting temperature; ventilation strategy

## 1. Introduction

With the development of urbanization and the improvement of people's living standard, the energy consumption of the building sector has been increasing rapidly in recent years. Energy consumption in the building operational sector accounts for one-third of the world's total energy consumption, and it will increase to nearly 50% by the year 2050 [1]. In China, the building sector is the second largest energy consumption sector in China's national economy [2], contributing about 31.4% of the total energy consumption [3]. Additionally, about 30–60% of total China's energy consumption in buildings is attributed to the space heating, ventilation and air conditioning (HVAC) according to the local climate [4–6]. Energy conservation is becoming more and more important because of the country's limited energy resources.

Building external envelopes are the bridges between indoor and the outdoor environment which affect the indoor heat gain and heat loss. Though the roof is a relatively smaller part of external building envelopes in terms of the surface area, the heat gain through the roof is considerable due to the high

intensity of solar radiation in summer. In the south of China, the summer is very hot and lasts for a long time. The cooling period lasts 4–8 months a year, and the heating period is very short, or even unnecessary. The maximum temperature of the outer surface can be above 70 °C (74 °C in Wuhan, 76 °C in Guangzhou) in summer. Such high roof surface temperature leads to excessively high inner surface temperature and high air conditioning load, which affects indoor thermal comfort. Therefore, improving roof thermal performance is crucial to reduce air conditioning energy consumption and improve indoor thermal comfort. The thermal performance of the roof has a large impact on the energy consumption of the air conditioning system.

To reduce the energy demand of buildings, the adoption of various passive energy saving techniques such as insulation material, green roof, night ventilation or other energy-efficient systems such as wind-catchers [7], atriums [8] and so forth is required. One of the energy-saving technologies for roofs mainly uses insulation material to increase the thermal resistance and reduce the thermal transfer caused by temperature difference [9,10]. However, research has found that thermal insulation has an extremely small effect on reduction of cooling energy consumption and a large effect on reduction of heating energy consumption. Taking Changsha, a capital city of a neighboring province, as an example, when the thermal insulation layer thickness increases from 0 to 100 mm, the cooling energy consumption first decreases and then gradually increases; the largest decrease is 1.04% compared with noninsulation envelope [11]. Green roof and night-time ventilation are also promising passive roof cooling strategies [12,13]. However, green roofs tend to be more expensive compared with normal roofs and require extra maintenance depending on the irrigation needs and vegetation type [14].

Recently, integrating phase change materials (PCMs) into roofs to increase their thermal mass has been recognized as an effective strategy for energy conservation of buildings. PCMs store energy in the form of latent heat by phase change process between solid and liquid. It can prevent overheating in summer, reduce the heat flux through the roof and reduce the energy consumption for space cooling [15,16]; furthermore, the use of PCM in the building envelope reduces indoor temperature fluctuations, thus providing better indoor thermal comfort for occupants [17]. So, integrating PCMs is a good method in southern China.

Many studies have been conducted on the application method of combining PCMs with envelops since the early 1990s. Methods of combining PCMs with envelop materials generally are divided into three types: (i) immersion, (ii) incorporation (iii) encapsulation. The immersion method is immersing porous materials of envelope in melted PCM, and incorporation is adding PCMs to supporting materials directly (gypsum board and concrete, et al.). Form-stable PCM can be produced using these two methods. The advantage of Form-stable PCM is that there is no macroscopic change in the state of the composite before and after phase transition, and no container is required. The mass ratio of PCMS to supporting materials should be controlled to prevent PCMS leaking from the supporting materials during melting [18]. Encapsulation method is packing PCMs in containers or microscopic capsules [19]. Microencapsulated PCM is wildly used in building materials [20]. Hawes and Feldman experimentally studied the thermal storage capacity of organic PCMs directly combined with concrete [21] and gypsum board [22] and found that the thermal storage capacity of PCM concrete is 10 times higher than that of ordinary wallboards [21]; the latent heat of PCM gypsum board is more than 180 kJ/kg [22]. Ahmet [23] used high-density polyethylene as the supporting material and two kinds of paraffin as the PCMs to prepare the form-stable PCM. The values of latent heat were 192.8 and 212.4 kJ/kg, respectively. Myriam [24] and Alexander [25] encapsulated PCMs and added them to the concrete and textile reinforced panels, respectively. The results showed that this structure can obtain a reduction and a time-shift in the maximum heat flux through the composite wall. Navarro [26] proposed a new constructer consisting of a prefabricated concrete slab with PCM macroencapsulated in small tubes and inserted in its hollows; the thermal inertia and heat storage capacity of roof have been improved. Some scholars have proposed to add graphite nanoplatelets [27] or expanded graphite [28] in PCMs to improve the thermal conductivity of form-stable PCMs.

The effect of the PCM envelopes on thermal performance of buildings has been extensively studied in the past. Pasupathy et al. [29] studied heat transfer characteristics of the PCM roof in Chennai of Italy using numerical model and concluded that double-layer form-stable PCMs could reduce the indoor temperature fluctuations significantly. Jin et al. [30] established a numerical model of a floor slab with two-layer form-stable PCM using finite deference method. The energy released by composite slab were 41.1% and 37.9% higher than that of the ordinary slab during heating and cooling, respectively. Tokuc et al. [31] established one-dimensional numerically heat transfer model of roof with PCM. Results showed that the optimum thickness of the PCM for flat roofs in Istanbul was 2 cm. Li et al. [32] investigated the effect PCM on the thermal performance of the roof by Fluent. Temperature reduction of 10–13 °C can be obtained in the inner surface temperature. Han [33] studied the energy saving characteristics of PCM-embedded building envelopes in four climates through building network simulation by EnergyPlus and found that HVAC system could save 17% energy per year. Saffari et al. [34] investigated the effects of PCMs with different melting temperatures and thicknesses in Madrid. The energy consumption of air conditioning system could be reduced by 10% to 15%. Xaman [35] simulated the thermal performance of a concrete roof with a form-stable PCM layer on its interior surface under a Mexican warm weather (Merida); results showed that 2 cm paraffin wax MG29-layer added in a concrete roof could diminish 57% of the thermal load. Tokuç [36] investigated the thermal performance of a roof with form-stable PCM in four cities of Turkey using Fluent. Two centimeters of PCM is placed in the middle layer; the results showed that the cooling loads were reduced by 48.2~99.1%. Dnyandip [37] studied the effect of inclination PCM integrated roof on cooling load reduction using ANSYS Fluent. Inorganic salt hydrate layer thickness of 2.5 cm with inclination angles of 0°, 2° and 4° were investigated. It was found that by placing an inclination PCM layer in the middle of the roof, the daily heat gain could be significantly reduced, and PCM layer at an angle of 2° demonstrated the best performance in Chennai. PCM roof also has a good energy saving effect in winter. Devaux and Farid [38] demonstrated the benefits of PCM when incorporated in walls and roofs for an experimental hut. Tolga Pirasaci [39] analyzed the winter energy saving potential of a residential apartment incorporated with a PCM layer; the results showed that annual heating energy requirement decreased by about 6%.

In summary, studies on PCM roof mainly focused on the application of PCMs combined with building materials, and some focused on the parametric analysis, optimal design and the thermal performance of PCM integrated roof. Form-stable PCMs in the middle and the interior layer of roof are wildly used and investigated. Studies found that PCM roofs can reduce temperature amplitude, delay the occurrence of peak temperature and reduce the heat transfer from the envelope. Although PCM is widely used in building envelopes, the energy saving potential of PCM roofs in summer is limited since the stored heat in PCM in the daytime will become a secondary heat source; the stored heat will be released and transferred to the room, which will lead to an increase in indoor temperature and cooling load at night.

Very few studies focused on the combination of shape-stabilized PCM and night ventilation on the roof. A new ventilated roof with form-stable PCM (VRFP) is proposed to removal the stored heat in PCM by night ventilation. The novelty of this structure is that both the PCM layer and the ventilation layer are designed. The prefabricated hollow slab is as the main structure of roof, and the hollow part of slab is as ventilation ducts. Mechanical ventilation is carried out at night by providing outdoor cold air to flow through the cavity. The heat gains from solar radiation, outdoor air and environment radiation can be stored in PCM during melting process in the daytime and released by PCM at night. The night ventilation can take away the heat generated during the PCM solidification, solve the problem that PCMS cannot completely solidify at night, improve the utilization of PCM latent heat and further reduce the heat transfer into the room. In addition, it can store the excess cooling energy of outdoor cool air by full use of the thermal storage capacity of PCM and concrete and greatly reduce the cooling load. The main contributions of this study are two points: First, ANSYS Fluent numerical simulation method was adopted, the key influence factors affecting the thermal performance of VRFP were explored,

the comprehensive effects of melting temperature, PCM thickness and ventilation strategy on thermal performance of VRFP were studied; the thermal performance of this roof was evaluated by analyzing inner surface temperature, latent heat utilization rate of PCM and decrement factor. Second, the most optimal melting temperature, the suitable PCM layer thickness, the optimal ventilation speed and time were determined in Wuhan. The results provide reference on optimization design of the structure and are of great significance for the application and promotion of the structure.

## 2. Construction and Materials of VRFP

The construction of VRFP is shown in Figure 1. The roof consists of protective layer, waterproof layer, form-stable PCM layer, leveling layer, hollow core slab and the interior plaster. The first layer is protective layer, which is made of the cement mortar; the thickness is 20 mm. The second layer is the waterproof layer; it is polymer modified asphalt waterproof rolling material of 2 mm thick. The third layer is form-stable PCM. The selection of PCM has four key principles: the appropriate melting temperature range; high latent heat, low cost, nontoxic and noncorrosiveness. Paraffin is cost-effective and has considerably high latent heat (150–250 kJ/kg) and suitable melting temperature [40], so the form-stable PCM is obtained by dispersing the paraffin into high density polyethylene (HDP) with the mass percentage of 85% through compression and drying technologies, in which high density polyethylene is support material. The HDP is used as supporting material. The latent heat of PCM is 200 kJ/kg. The fourth layer is a 20 mm leveling layer, and the material used is cement mortar. The fifth layer is concrete structure with air duct embedded. Precast hollow core slab is selected here; the cavities are used for ventilation duct. The fan supplies outdoor cool air flowing through the cavity at night and removal the heat released by PCM solidification process. The thickness of the precast hollow core slab is 120 mm, and the cavity diameter is 80 mm. Thermal physical properties of materials in each layer of the VRFP are shown in Table 1.

The heat transfer mechanism is as follows:

1.  During summer daytime, the temperature of the roof increases under the effects of solar radiation, air convection heat transfer and environmental long-wave radiation. When PCM reaches its melting temperature, the state changes from solid to liquid, and meanwhile absorbs large amounts of heat gained from roof surface. The temperature of the PCM layer is basically unchanged during the phase transition process, so the inner surface temperature is kept lower in the daytime and the influence of solar radiation and outdoor temperature fluctuation is weakened.

2.  During summer night, the roof surface temperature drops under the effect of sky radiation and convection heat transfer. When the temperature of PCM drops to the solidification temperature, PCM gradually solidifies and releases the heat again. The ventilation begins when the outdoor air temperature drops below the roof temperature. Specifically, the ventilation begins when the temperature of bottom point at the middle length of the cavity is higher than outdoor air temperature and finishes when the temperature of bottom point at the middle length of the cavity is lower. So, the heat released by the PCM is removed by the ventilation; the cooling load can be reduced obviously.

This structure can avoid overheating phenomenon effectively and has advantageous performance on heat preservation and insulation. The goal in implementing ventilated PCM-roof is to significantly reduce and time-shift the peak cooling load in order to reduce the energy consumption for space cooling.

The thermal performance of VRFP in Wuhan is analyzed. It is located at inland at 30 degrees' north latitude and has a subtropical monsoon humid climate. Summer in Wuhan lasts for about 4–5 months. July is the hottest month, whose average temperature is 28.7 °C, and the total solar radiation in this month is up to 545.2 MJ/m$^2$. The number of hours above 35 °C is 101 h, and the maximum temperature that occurred is up to 44.5 °C. It is well known as one of the three furnace cities in China.

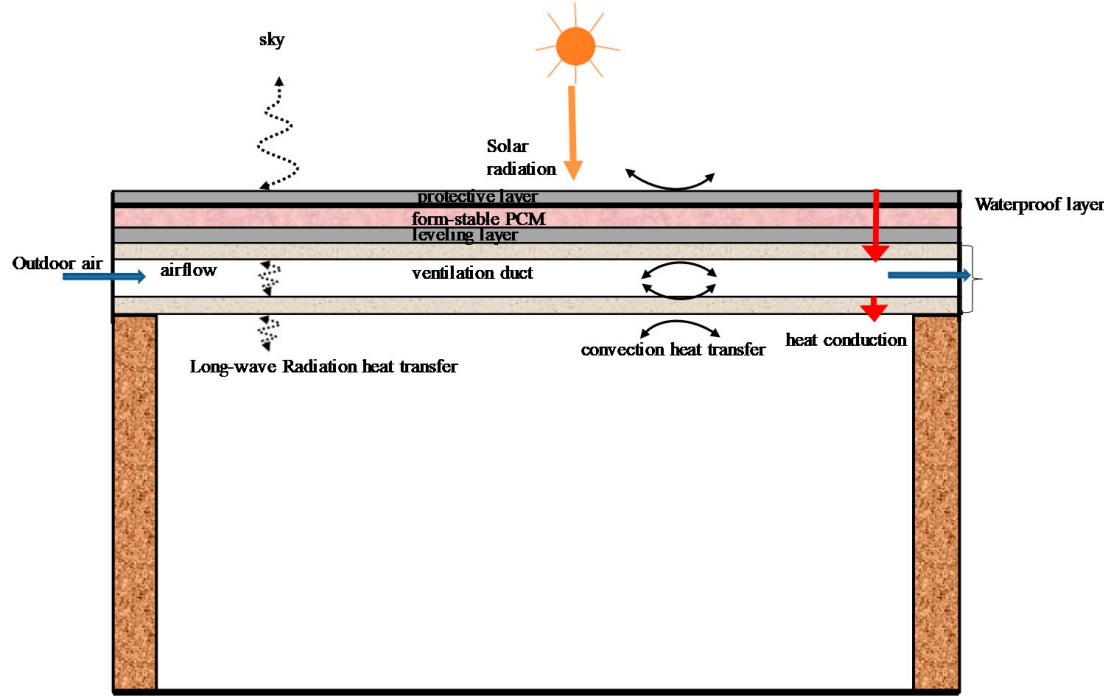

**Figure 1.** Ventilated roof with form-stable PCM (VRFP) structure.

**Table 1.** Thermal physical properties of materials in each layer of the VRFP.

| Structure | Material | Thickness (mm) | Density (kg/m$^2$) | Specific Heat (J/kg·K) | Thermal Conductivity (W/m$^2$·K) |
|---|---|---|---|---|---|
| Protective layer | Cement mortar | 20 | 1800 | 1050 | 0.93 |
| PCM layer | Paraffin/HDP | 30 | 755/940 | 2510/2100 | 0.2/0.5 |
| Screed-coat | Cement Mortar | 20 | 1800 | 1050 | 0.93 |
| Hollow slab | Concrete | 120 | 2500 | 920 | 1.7 |
| Hollow core | Air | D80 | 1.18 | 1005 | 0.02638 |

## 3. Methodology

Computation Fluid Dynamics (CFD) is a reliable numerical analysis method. It is usually used to analyze the heat transfer in phase change process of PCM and heat transfer of ventilated facade. In this study, the CFD package of ANSYS Fluent is used to investigate the thermal performance of VRFP. The effects of the melting temperature, the thickness of PCM layer, the ventilation speed and time on thermal performance of VRFP are analyzed in Wuhan, and the optimal design of VRFP is obtained.

### 3.1. CFD Model

The cross section of the VRFP is shown in Figure 2. In the precast hollow core slab, the cavities are precast side by side. The fan supplies outdoor cool air flowing through each cavity, so one unit in the dotted line is selected to be studied. The model is shown in Figure 3. The length of a precast hollow core slab is 4.2 m. As the 2 mm waterproof layer has little influence on heat transfer, it was ignored in CFD numerical simulation for the purpose of simplifying the model. The commercial CFD package of ANSYS Fluent 14.5.7 was employed to study the thermal performance of the structure. A three-dimension model was established. The standard $k$–$\varepsilon$ is suggested to model the heat transfer and airflow process of the ventilation, especially for the turbulence [41–43]. The standard $k$–$\varepsilon$ model, DTRM radiation model and solidification-melting model were used. The SIMPLE scheme for velocity–pressure coupling was applied to solve PCM momentum equations. The under-relaxation factors of 0.9 were selected to improve the convergence stability.

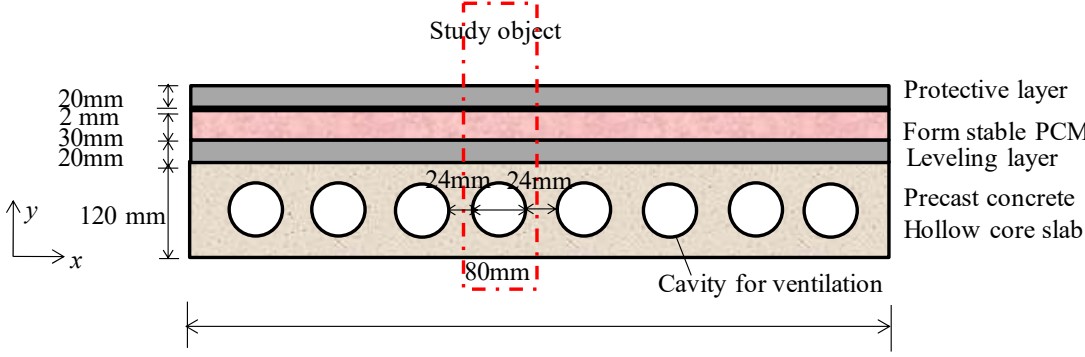

**Figure 2.** The cross section model of VRFP.

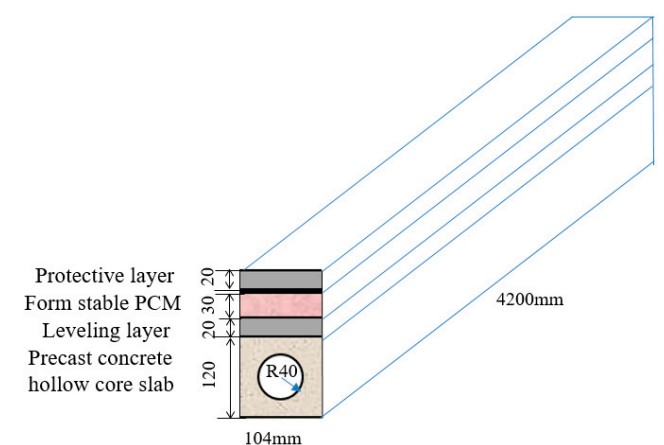

**Figure 3.** The model of VRFP.

### 3.2. Governing Equations and Boundary Conditions

The heat transfer of the roof is calculated in five parts; they are protective layer, PCM layer, leveling layer, a hollow slab as well as the ventilation air

(1) when the materials are protective layer, leveling layer and hollow slab layer

The unsteady energy equation of the leveling layer and hollow slab is presented as below:

$$\rho_i c_{pi} \frac{\partial T}{\partial \tau} = \lambda_i \left( \frac{\partial^2 T}{\partial x^2} + \frac{\partial^2 T}{\partial y^2} + \frac{\partial^2 T}{\partial z^2} \right) \tag{1}$$

where $\tau$ is time (s). $T$ is temperature (K). $\rho$, $c_p$ and $\lambda$ are density (kg/m$^3$), specific heat at constant pressure (J/(kg·K)) and thermal conductivity (W/(m·K)), respectively. Subscript $i$ represents the $i$th layer.

(2) When the material is PCM

A solidification-melting model of PCM can calculate the heat transfer during the solidification and melting process. The energy transfer equation of the solid region is shown as follows:

$$\frac{\partial}{\partial \tau}(\rho h) + \nabla \cdot \left( \vec{v} \rho h \right) = \nabla \cdot (\lambda \nabla T) + S_h \tag{2}$$

where, $h$ is sensible enthalpy (J/kg), $h = h_{ref} + \int_{T_{ref}}^{T} c_p dT$; $h_{ref}$ is reference enthalpy; $T_{ref}$ is reference temperature; $\vec{v}$ is fluid velocity (m/s); $S_h$ is volumetric heat source (J/m$^3$);

The enthalpy $H$ in the energy equation of the phase change region differs from that of the solid region. The calculation formula is as follows:

$$H = h + \Delta H = h + \beta L \tag{3}$$

where, $\Delta H$ is latent enthalpy (J/kg). $L$ is the latent heat of the PCM. $\beta$ is liquid fraction, which is defined as follows:

$$\begin{cases} if \ T < T_{solid}, & \beta = 0 \\ if \ T_{solid} < T < T_{liquid}, & \beta = \frac{T - T_{solid}}{T_{liquid} - T_{solid}} \\ if \ T > T_{liquid}, & \beta = 1 \end{cases} \tag{4}$$

$T_{solid}$, $T_{liquid}$-the solid temperature and liquid temperature, respectively (K).

Instead of tracking the liquid-solid front explicitly, ANSYS Fluent uses an enthalpy-porosity formulation. For the solidification-melting process, the energy equation is written as:

$$\frac{\partial}{\partial \tau}(\rho H) + \nabla \cdot \left(\vec{v} \rho H\right) = \nabla \cdot (\lambda \nabla T) + S_h \tag{5}$$

The solution for temperature is essentially an iteration between the energy equation and the liquid fraction equation.

(1)   When the material is air but the air does not flow

$$\rho_a c_a \frac{\partial T}{\partial \tau} = \lambda_a \left( \frac{\partial^2 T}{\partial x^2} + \frac{\partial^2 T}{\partial y^2} + \frac{\partial^2 T}{\partial z^2} \right) \tag{6}$$

(2)   When the air flows for ventilation

The model used for calculating the unsteady heat transfer and airflow in the ventilated cavity is list as below. Continuity equation is list as below shown as:

$$\frac{\partial \rho_a}{\partial \tau} + \nabla \left( \rho_a \vec{v} \right) = 0 \tag{7}$$

where subscript $a$ represents the air and conservation of momentum is provided as below:

$$\frac{\partial \left( \rho_a \vec{v} \right)}{\partial \tau} + \nabla (\rho_a \vec{v} \vec{v}) = -\nabla P + \nabla (\vec{f}) + \rho_a \vec{g} + \vec{F} \tag{8}$$

where $P$ is the static pressure. $\vec{f}$ represents the shear stress tensor. $\rho_a \vec{g}$ represents the body force. $\vec{F}$ represents the external force. Conservation of energy is given as follows.

$$\frac{\partial (\rho_a E)}{\partial \tau} + \nabla \vec{v} (\rho_a E + P) = -\nabla \left[ \lambda_{eff} \nabla (T) - \sum_i h_i \vec{J_i} + \left( \vec{f_{eff}} \vec{v} \right) \right] + S^T \tag{9}$$

where $\lambda_{eff}$ represents the effective thermal conductivity; $h_i$ is convective heat transfer coefficient; $J_i$ is diffusion flux of species. $f_{eff}$ is effective shear stress tensor. No-slip conditions were applied on the surfaces of the ventilated gap.

### 3.3. Boundary Conditions and Gird Sensitivity Analysis

The simulation model of VRFP is shown in Figure 3; the boundary conditions are as follows:

(1) The left and right sides of the object unit were both set as adiabatic boundary conditions. (2) The up side and bottom side surfaces of the unit were adopted the third boundary conditions. The convective heat transfer coefficients of the outer and inner surfaces of the roof were 23.26 W/m$^2$ and 8.72 W/m$^2$, respectively [44]. The indoor air temperature was set at 26 °C. The outdoor solar-air temperature was used as the external boundary condition. (3) When the roof is ventilated at night, the inlet air velocity and temperature are the input parameters, and the exit is free flow.

The outdoor solar-air temperature includes the effects of outdoor air temperature, solar radiation and long-wave radiation of environment and was used for the boundary of outdoor environment. The calculation formula is as follows:

$$t_z = t_a + \frac{\rho \cdot I}{\alpha_{out}} - \frac{Q_l}{\alpha_{out}} \tag{10}$$

where, $t_z$ is the solar-air temperature, $t_a$ is the outdoor air temperature, $\rho$ is the solar radiation absorption coefficient, here mid-color is assumed and $\rho$ is 0.75; $I$ is the intensity of solar radiation, W/m$^2$; $\alpha_{out}$ is the outdoor convective heat transfer coefficient, here is 23.2 W/m$^2$·K; $Q_l$ is the long-wave radiative radiation between the roof and the sky, $\frac{Q_l}{\alpha_{out}}$ is 4 °C [45], so the solar-air temperature is calculated and shown in Figure 4 according to the intensity of solar radiation and the outdoor air temperature in summer typical day.

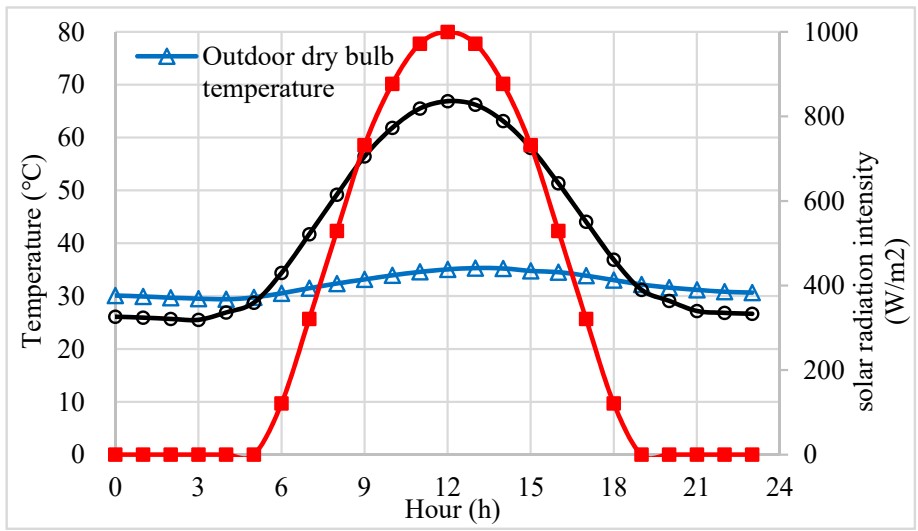

**Figure 4.** The solar-air temperature on a typical summer day in Wuhan.

Hexagonal structure grid was used in the roof construction and O-block grid was used in the ventilated cavity when calculating the airflows and the energy fluxes. A local mesh refinement is adopted around the cavity. The mesh volumes vary from 32.5 mm$^3$ to 740.3 mm$^3$. During the simulation, the time step was 60 min, and the number of steps were set to 168. In order to perform the grid independence analysis, the inner surface temperature of the roof was calculated by Fluent model with grids of 20,000 cells, 250,000 cells and 640,000 cells, respectively. The details of CFD model establishment and the calculation results of inner surface temperatures under different grid numbers are displayed in Supplementary Materials F1. The results of the three girds were found to be basically the same with a less than 0.3% difference. The data of 250,000 and 640,000 gird numbers mostly coincide, indicating that the number of grids of 250,000 is accurate enough. Thus, the model with a grid of 250,000 cells was selected to simulate the VRFP model for accuracy and simplicity.

### 3.4. Accuracy Analysis of the CFD Model

The CFD analysis was developed using the ANSYS Fluent software, which solves the governing equations through the finite volume method. It can be used as an effective tool to solve the heat transfer problems in ventilation process. A large number of experiments verified the accuracy of CFD model in ventilation study. Buratti et al. [46] investigated the thermal performance of a new ventilated brick wall by means of experiment and CFD simulations. Three turbulent models (standard $k$–$\varepsilon$, standard $k$–$\omega$, laminar) were used for the simulation within the air gap, and results show that standard $k$–$\varepsilon$ model could obtain reliable results and was selected for the thermal performance investigation.

Diarce et al. [47] used the Fluent CFD code to study the thermal behavior of a new ventilated wall with outerlayer PCM, the RNG *k–ε* model which is based on the Standard *k–ε* model showed good agreement between the experimental and can be considered suitable for the simulation under turbulent flow conditions. In addition, many scholars used standard *k–ε* model to perform a CFD numerical study on the thermal performance of ventilated wall or roof [48–50]. From the above studies, we can conclude that CFD can be used to study the heat transfer of ventilation process with reliable results, and standard *k–ε* model was selected in this study.

A solidification-melting model included in Fluent can calculate the heat transfer during the solidification and melting process. Fluent CFD can be used to solve the heat transfer problems involving solidification and/or melting taking place at one temperature or over a range of temperatures. In order to validate the accuracy of solidification-melting model in CFD, the previous study [51] performed model accuracy analysis of heat transfers for an outerlayer form-stable PCM roof. The inner surface temperature of this PCM roof was simulated and compared with the experimental data [52]; results showed that the maximum relative errors of simulated results are lower than 5%, indicating that the solidification-melting model in CFD are accurate and reliable. The accuracy of solidification-melting model in CFD for calculating the heat transfer of PCM roof is verified.

## 4. Results and Analysis

### 4.1. Thermal Performance Evaluation Index

Time lag and decrement factor are selected to evaluate the thermal inertia of the roof. The time lag refers to the time difference between the occurrence time of maximum inner surface temperature and the occurrence time of maximum outdoor solar air temperature. The longer the time lag is, the better the thermal inertia of roof is.

Decrement factor represents the attenuation of the inner surface temperature; the equation can be written as:

$$DF = \frac{A_i}{A_o} \tag{11}$$

where $A_i$ is the temperature amplitude of the roof's inner surface and $A_o$ is the temperature amplitude of the roof's outer surface. Besides time lag and decrement factor, the temperature of the roof's inner surface determines the heat gain through the roof, so it is also an important index to evaluate the thermal performance of VRFP.

Liquid fraction and utilization rate of latent heat are used to evaluate the utilization of the PCM and determine the thickness of PCM. Liquid fraction *β* is defined as Equation (4). It can reflect the degree of solidification or melting of PCM. To investigate the application of PCM, every 5 mm is one sublayer, and PCM is divided into six sublayers. The utilization rate of latent heat in a certain sublayer $UR_i$ is calculated as follows:

$$UR_i = \beta_i^{max} - \beta_i^{min} \tag{12}$$

where $\beta_i^{max}$ and $\beta_i^{min}$ are the maximum and minimum liquid fractions of PCM during a day in a certain sublayer; the average utilization rate of latent heat of PCM *UR* is the mean value of $UR_i$.

### 4.2. The Optimal Melting Temperature and Suitable Thickness of PCM Layer

Melting temperature of PCM is often a range. The temperature between the solid temperature and liquid temperature (see Equation (4) for the definition) is the melting temperature range. The melting temperature range is recognized the same as the solidification temperature range. The melting temperature range of PCM determines when the phase transition begins and finishes and finally affects the roof's thermal behavior.

If the melting temperature is relatively high, the diurnal heat storage capacity of PCM is small, and the effect of PCM on the temperature attenuation of the roof's inner surface is not obvious; if the melting temperature is relatively low, PCM cannot fully solidify at night, and the available PCM is

reduced in the daytime, which also results in the low diurnal heat storage capacity. So, there is an optimal melting temperature range. When the roof shows the best thermal performance compared with the roofs to other melting temperature ranges, that is, the reduction in peak temperature of the roof's inner surface is the largest and the temperature amplitude of the roof's inner surface as well as the decrement factor of the building roof is the lowest, PCM is in the optimal melting temperature range.

The thickness of PCM layer affects the initial investment and the thermal insulation effect of roof. The thicker the PCM layer is, the better the heat insulation effect becomes and the lower the latent heat utilization rate is. When the thickness exceeds a certain level, increasing the thickness makes little improvement in insulation. The optimum thickness of PCM is expected to make initial investment lower and utilization of PCM higher, which ensures that the PCM has sufficient heat storage capacity. Therefore, the determination of the optimal thickness of PCM layer depends on the latent heat utilization rate of the PCM. If the latent heat utilization ratio is relatively low, the PCM layer is too thick; if the utilization ratio is 100%, the thickness may be insufficient; when the latent heat utilization ratio is lower than 100% and the value is large enough (more than 90%). This is considered to be the best thickness. A suitable thickness range is suggested considering the roof's inner surface temperature, decrement factor and latent heat utilization rate.

To investigate the effects of the melting temperature and the thickness of PCM layer on the thermal performance of VRFP, the thermal performance of this roof was simulated in Wuhan under nonventilation conditions. According to the previous research studies, the melting temperatures used in the experiments and simulations are between 22 and 41 °C [51,53–55], and the thicknesses of PCM layer are between 15 and 70 mm [29,31,36,37,51,54]. The proper ranges are related to the position of PCM in the structure and the climate of building location. The closer the phase change material is to the outside, the higher the melting temperature and the thicker the PCM required. So here the melting temperature range of 34–39 °C and the thickness range of 20–70 mm have been studied.

The relationship between decrement factor of VRFP and thickness of PCM layer under different melting temperature ranges is shown in Figure 5. For the melting temperature range of 36~38 °C or 37~39 °C, when the PCM thickness increases from 20 mm to 30 mm, there shows a large reduction in decrement factor. When thickness increases to 40~70 mm, the decrement factor has a small reduction and closes to 0.03 and 0.045 respectively. Therefore, the suitable thickness is 30 mm when the melting temperature is 36~38 °C and 37~39 °C considering the effect on the decrement factor. As for the melting temperature of 35~37 °C, the decrement factor is greatly reduced when the thickness increases from 20 mm to 40 mm, and the decrement becomes much slower when the thickness increases from 50 mm to 70 mm, and gradually approaches to 0.21, so that the suitable thickness is 40~50 mm when the melting temperature is 35~37 °C. When the melting temperature is 34~36 °C, the decline in decrement factor is very fast even if the thickness increases to 70 mm, which indicates that VRFP with lower melting temperature needs much thicker PCM layer.

The relation between the latent heat utilization rate and thickness of PCM layer under different melting temperature ranges is shown in Figure 6. The variation of latent heat utilization is roughly the same at different melting temperature ranges. The figure shows that the average latent heat utilization rate with melting temperature ranges of 35–37 and 36~38 °C are always the larger in the case of different PCM thicknesses. The latent heat utilization rate decreases with the increase of the thickness, and the rate of decline gradually decreases. When the thickness is 20 mm, the latent heat utilization rates are all 1.0 regardless of the phase transition temperature, which means 20 mm PCM is too thin; when the thickness is 30~40 mm, the latent heat utilization rates are high and the values vary from 0.60 to 0.96 in terms of different phase transition temperatures. When the thickness is larger than 50 mm, the latent heat utilization rates are much lower and the thickness is too thick that part of the PCM is useless. The results show that when the thickness of PCM layer is 30~40 mm, the latent heat utilization ratio of the roof is high and the thickness is appropriate.

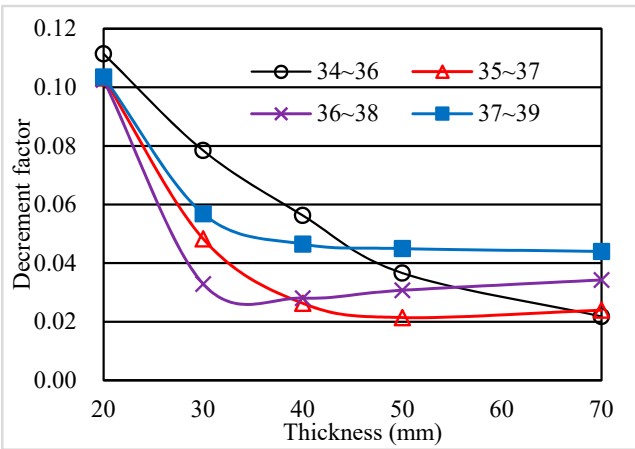

**Figure 5.** The relation between decrement factor and thickness of phase change material (PCM) layer.

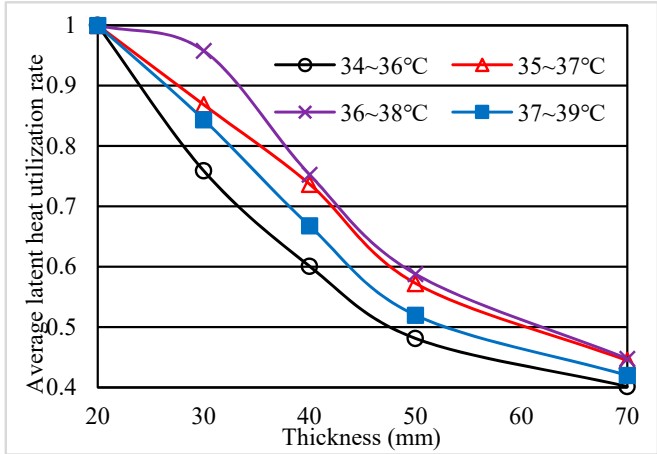

**Figure 6.** The relation between average utilization rate of latent heat and thickness of PCM layer.

The optimal melting temperature range at each PCM thickness is obtained by a comprehensive comparison of average temperature of the roof's inner surface, utilization rate of latent heat and decrement factor. As shown in Table 2, compared with the case without the PCM, with the increase of PCM thickness, the average temperature of inner surface and decrement factor decreases, time lags of roof increase 2–3 h in terms of different PCM thicknesses. It is concluded that the thermal properties of the roof have a better performance in all aspects when the thickness of PCM layer is 30~40 mm; the optimal melting temperature ranges are 36~38 °C.

**Table 2.** Optimal melting temperature ranges and thermal performance at different PCM thicknesses.

| PCM Thickness (mm) | Optimal Melting Temperature (°C) | Average Temperature of Inner Surface (°C) | Decrement Factor | Time Lag (h) | Average Latent Heat Utilization Rate |
|---|---|---|---|---|---|
| 0 | — | 30.98 | 0.232 | 5 | — |
| 20 | 36~38 | 30.62 | 0.103 | 7 | 100% |
| 30 | 36~38 | 30.43 | 0.033 | 8 | 96% |
| 40 | 35~37 | 30.28 | 0.026 | 8 | 74% |
| 50 | 35~37 | 30.09 | 0.021 | 7 | 57% |
| 70 | 34~36 | 29.90 | 0.022 | 7 | 44% |

The hourly inner surface temperature was obtained by simulating the roof with phase change layer thickness of 30 mm. Figure 7 shows the hourly temperatures of the inner surface of VRFP with and without PCM layer. For conventional roof without PCM, the average inner surface temperature of

the roof is 30.98 °C, the peak temperature is 34.71 °C, the temperature amplitude is 7.01 °C, the time lag is 5 h and the decrement factor is 0.232. When the thickness of PCM layer is fixed, the melting temperature has a weak influence on the average temperature of the roof's inner surface but has a great influence on peak temperature, temperature amplitude and decrement factor of the roof's inner surface. When the melting temperature is 36–38 °C, the inner surface temperature varies from 30.0 to 30.8 °C; the temperature amplitude is 0.80 °C and it is the lowest among the different melting temperature ranges. Compared with the roof without PCM, the peak temperature can be reduced by 4 °C.

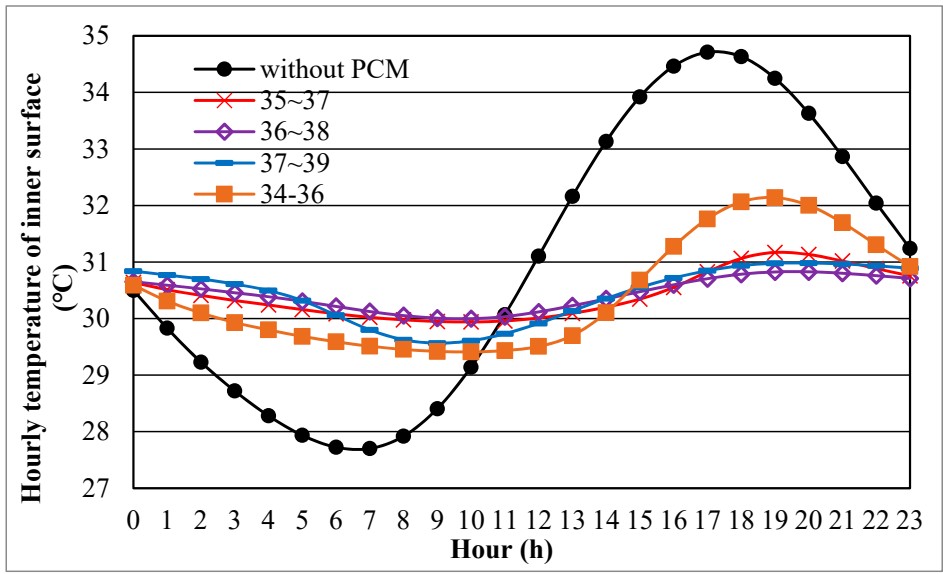

**Figure 7.** Hourly temperature of the roof's inner surface with a 30 mm PCM layer.

### 4.3. Ventilation Speed and Ventilation Time

On summer nights, roof temperature drops under the effects of sky radiation and convective heat transfer of outdoor cold air and the PCM solidifies. In order to prevent the heat released by the PCM transferring into the room, the cavity of hollow slab is cooled by outdoor cool air at summer night through mechanical ventilation. It solves the problem of low utilization of latent heat because the solidification heat at night cannot be eliminated in hot weather. In addition, the free cooling energy brought by outdoor air can be stored in the PCM layer and the roof structure. Therefore, the ventilation reduces the temperature of the roof's inner surface and increases the time lag.

A real-time monitoring ventilation strategy was used to control the ventilation time and ventilation speed of the roof panel. In order to ensure the cooling effect of ventilation, the temperature of the bottom point at the middle length of the cavity is monitored. If the outdoor dry-bulb temperature is lower than monitoring point temperature, the outdoor air will be introduced by fan into the cavity for removing the heat. When the outdoor air temperature is higher than this point, the ventilation stops. According to analysis in previous section, the suitable thickness is 30~40 mm and optimal melting temperature ranges are 35~38 °C in Wuhan. So, the VRFP with melting temperature range of 35~37 °C and PCM layer thickness of 30 mm was selected as the research object in ventilation process to study the effect of ventilation strategy on the thermal performance of VRFP.

With the increase of the ventilation speed, the temperature of the roof's inner surface decreases, while the electric consumption of the fan and the noise of ventilation duct increase. Considering the above factors, the ventilation speed range of 0–2.4 m/s are studied. In the typical summer day of Wuhan, VRFP at ventilation speeds of 0 m/s, 0.4 m/s, 0.8 m/s, 1.2 m/s, 2.0 m/s and 2.4 m/s were simulated and the thermal characteristics of VRFP were obtained. Hourly temperatures of the roof's inner surface are shown in Figure 8. The thermal performance of VRFP is shown in Table 3. Compared with the roof without ventilation, when the VRFP is ventilated by the outdoor air, the minimum temperature of

the roof's inner surface is significantly reduced at night and the maximum temperature of the roof's inner surface is decreased apparently during the daytime. With the increase of ventilation speed, the maximum temperature changes little, while the minimum temperature decreases significantly; temperature amplitude and decrement factor firstly increase and then decrease.

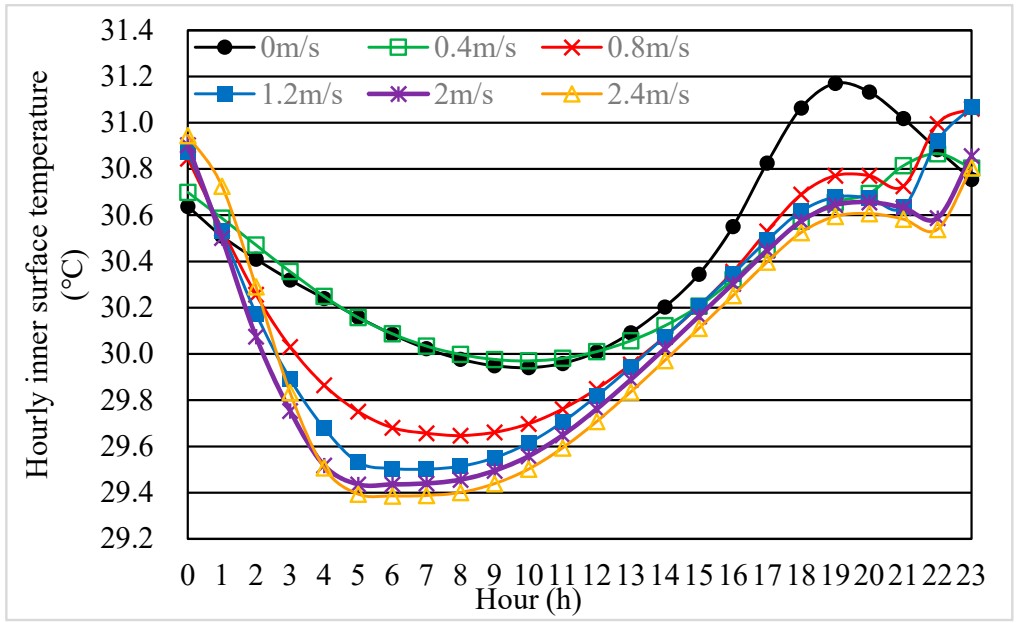

**Figure 8.** Hourly temperatures of VRFP inner surface at different ventilation speeds.

**Table 3.** The thermal performance of VRFP at different ventilation speeds.

| Evaluation Index | Without PCM and Ventilation | Ventilation Speeds | | | | | |
|---|---|---|---|---|---|---|---|
| | | 0 m/s | 0.4 m/s | 0.8 m/s | 1.2 m/s | 2.0 m/s | 2.4 m/s |
| Average temperature of inner surface (°C) | 30.98 | 30.49 | 30.34 | 30.22 | 30.15 | 30.07 | 30.05 |
| Peak temperature of inner surface (°C) | 34.71 | 31.17 | 30.87 | 31.06 | 31.07 | 30.96 | 30.60 |
| Minimum temperature of inner surface (°C) | 27.70 | 29.97 | 29.97 | 29.65 | 29.50 | 29.42 | 29.39 |
| Temperature amplitude of inner surface (°C) | 7.01 | 1.23 | 0.90 | 1.41 | 1.57 | 1.54 | 1.21 |
| Decrement factor | 0.232 | 0.048 | 0.035 | 0.055 | 0.061 | 0.060 | 0.047 |
| Time lag (h) | 5 | 7 | 9 | 11 | 11 | 11 | 12 |

Compared with nonventilation condition, when the ventilation speed is 0.4 m/s, the highest temperature of the roof's inner surface during 14:00~20:00 is reduced by about 0.5 °C. That is because the outdoor low-temperature air at night takes away the heat released by PCM and stores the cold energy in the roof and makes the effect of heat absorption during daytime better. When the ventilation speed at night is increased to more than 0.8 m/s, the temperature of the roof's inner surface at night is significantly reduced. The higher the ventilation speed is, the more significant the cooling effect is. When the ventilation speed is 2.4 m/s, the highest and lowest temperatures of inner surface temperature are both decreased by 0.6 °C. The maximum temperature is delayed from 6 p.m. to 9~10 p.m. The time lag is increased from 7 h to 9~12 h. Ventilation is very helpful for the improvement of the indoor thermal comfort.

It can be seen from Figure 8 that when the cavity begins to be ventilated, the temperature of the roof's inner surface will suddenly increase within 1 or 2 h and reach the maximum temperature. Therefore, the maximum temperature generally appears at 10–11 p.m. This is because at the beginning

of ventilation, the outdoor air temperature is only slightly lower than the temperature at the measuring point. After the outdoor air is introduced into the cavity, its temperature rises rapidly. The airflow absorbs heat in the first half of the cavity but heats the second half. As the ventilation time increases, the temperature of the roof's inner surface gradually decreases. The higher the ventilation speed is, the larger the inner surface temperature decreases.

The ventilation strategies for different ventilation speeds at the inlet of the cavity are shown in Figure 9. Ventilation strategies indicate the start and stop times of the ventilation for each air speed. When the ventilation speeds are 0.4 m/s and 0.8 m/s, ventilation starts at 10 p.m. and lasts for 8 h. Similarly, when the ventilation speed is 1.2 m/s, ventilation starts at 10 p.m. and lasts for 7 h. When the ventilation speeds are 2.0 m/s and 2.4 m/s, ventilation starts at 11 p.m. and lasts for 6 h.

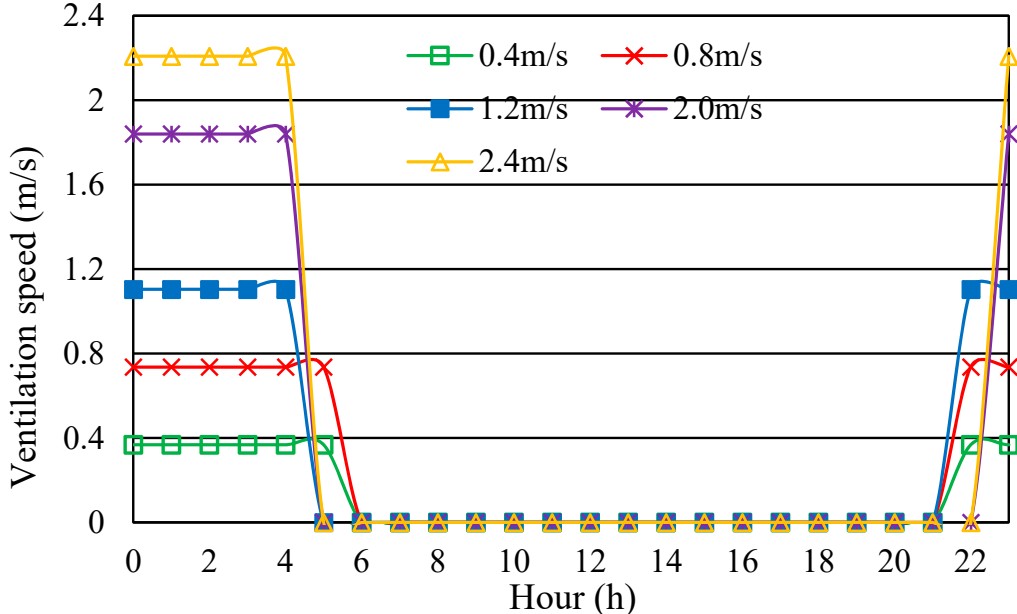

**Figure 9.** Hourly ventilation strategies at the inlet of the cavity.

Figures 8 and 9 indicate that the ventilation is stopped at 5 a.m. or 6 a.m., and then the temperature of the roof's inner surface begins to increase gradually. Further, the rising rate of the inner surface temperature of the ventilated roof is greater than that of the roof without ventilation, since there is a large temperature difference between the inner surface temperature and the outdoor solar-air temperature. However, the temperature rise of the inner surface of the ventilated roof is significantly less than that of the roof without ventilation at 4 p.m. to 9 p.m. This is because the heat exchange between the outdoor low-temperature air and the cavity takes away heat released by PCM and stores the cooling energy during nighttime ventilation.

The purpose of ventilation is to remove the heat during the PCM solidification and reduce the heat transfer through the roof. The optimal ventilation speed is expected to obtain the minimum heat transfer through the roof. The heat transfer through the roof is equal to the difference between the temperature of the roof's inner surface and the indoor air temperature multiplied by the convective heat transfer coefficient of the roof's inner surface. Because the convective heat transfer coefficient and the indoor air temperature are constants in this study, the temperature of the roof's inner surface directly decides the amount of heat gain through the roof. The relationship between the average temperature of the roof's inner surface and the ventilation speed is shown in Figure 10. With the increase of ventilation speed, the average temperature of the roof's inner surface decreases, but the reduction gradually becomes weaker. The optimum ventilation speed is obtained by finding the lowest average inner surface temperature of the roof.

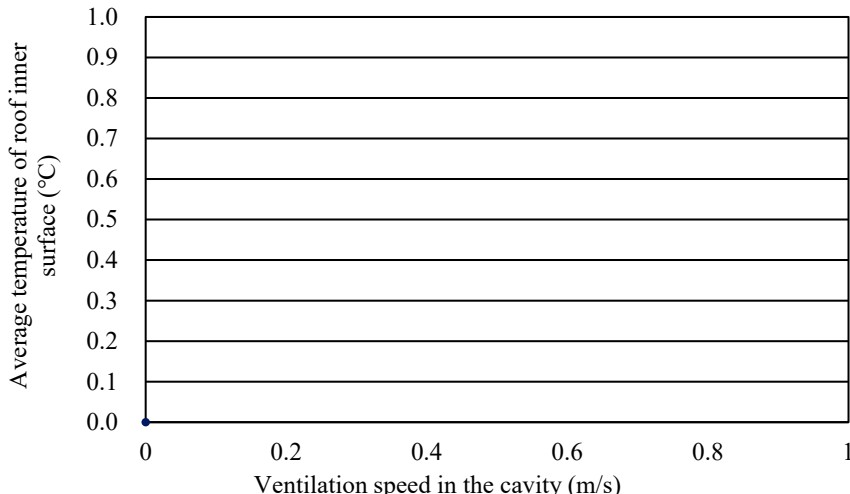

**Figure 10.** The relation between average temperature of the roof's inner surface and ventilation speed.

The fitting formula between the airflow rate and the average temperature of the roof's inner surface is obtained (Figure 10). As the ventilation speed increases, the average temperature decreases slowly. When the ventilation speed exceeds 2 m/s, the decrease rate of the average temperature is very slight. The optimal ventilation speed is the airflow velocity at which the reduction rate in the average temperature of inner surface is zero. It can be calculated from the fitting formula that the reduction rate of the average temperature gradually approaches to zero when the speed is 2.4~2.5 m/s. Therefore, the optimal ventilation speed range is 2.4~2.5 m/s in Wuhan.

To analyze the influence of ventilation on phase change process, hourly liquid fraction and latent heat utilization rate of each PCM sublayer without ventilation and with a ventilation speed of 0.4 m/s, 0.8 m/s, 1.2 m/s, 2.0 m/s and 2.4 m/s were studied. The results are shown in Figure 11 and Table 4.

The PCM layer with a thickness of 30 mm was divided into six sublayers from top to bottom (5 mm per sublayer) for detailed research. The latent heat utilization rate of each layer at different ventilation speeds is shown in Table 4. Figure 11a is the hourly liquid fraction and latent heat utilization rate of PCM under the condition of nonventilation. The PCM is completely melted during the daytime, but the PCM layer of 10–15 mm, 15~20 mm, 20~25 mm and 25~30 mm cannot be entirely solidified without ventilation at night. After the cavity is ventilated, the minimum liquid faction of each layer is gradually reduced. Because the PCM stores cooling energy from the outdoor low-temperature air at night. Further, latent heat utilization rates of last four layers are gradually increased with the increase of the ventilation speed. After the ventilation speed is increased to 1.2 m/s, PCM layer of 10~15 mm, 20~25 mm and 25~30 mm can be totally solidified at night. The latent heat utilization rate of 15~20 mm layer grows slowly with the continuous increase of ventilation speed. The average utilization rate of latent heat of six sublayers tends to be 99% and remains stable when the ventilation speed of cavity is increased to more than 1.2 m/s.

**Table 4.** The latent heat utilization rate of PCM layer at different ventilation speeds.

| Ventilation Speed (m/s) | PCM Layer (mm) | | | | | | Average Utilization Rate of Latent Heat (%) |
|---|---|---|---|---|---|---|---|
| | 0~5 | 5~10 | 10~15 | 15~20 | 20~25 | 25~30 | |
| Without ventilation | 100% | 100% | 79% | 70% | 77% | 95% | 86.83% |
| 0.4 | 100% | 100% | 91% | 75% | 81% | 98% | 90.76% |
| 0.8 | 100% | 100% | 97% | 84% | 93% | 100% | 95.71% |
| 1.2 | 100% | 100% | 100% | 91.40% | 100% | 100% | 98.57% |
| 2.0 | 100% | 100% | 100% | 92.95% | 100% | 100% | 98.82% |
| 2.4 | 100% | 100% | 100% | 93.56% | 100% | 100% | 98.93% |

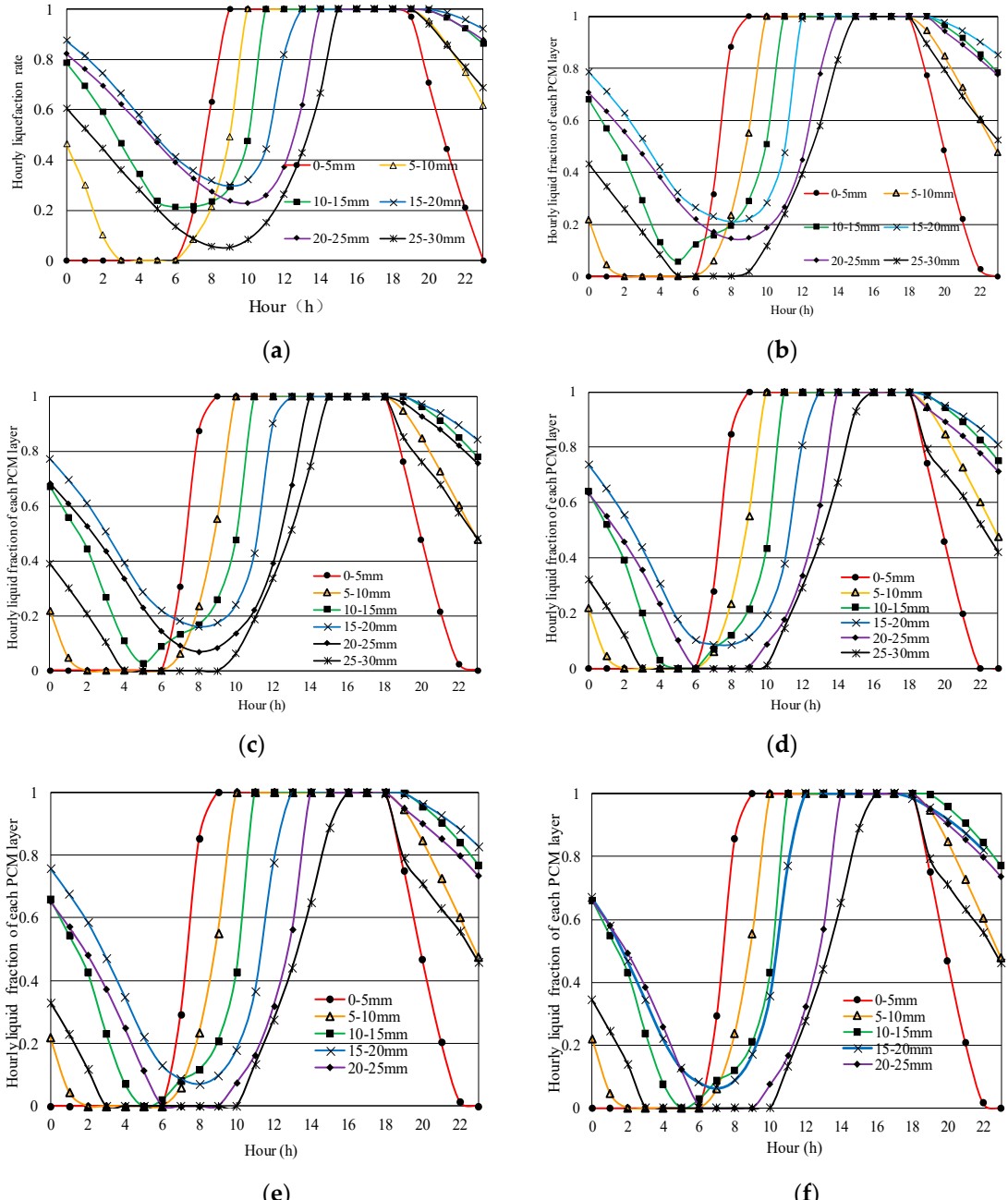

**Figure 11.** Hourly liquid faction of each PCM layer at different ventilation speeds. (**a**) Hourly liquid fraction of each PCM layer without ventilation; (**b**) Hourly liquid fraction of each PCM layer with a ventilation speed of 0.4 m/s; (**c**) Hourly liquid fraction of each PCM layer with a ventilation speed of 0.8 m/s; (**d**) Hourly liquid fraction of each PCM layer with a ventilation speed of 1.2 m/s; (**e**) Hourly liquid fraction of each PCM layer with a ventilation speed of 2.0 m/s; (**f**) Hourly liquid fraction of each PCM layer with a ventilation speed of 2.4 m/s.

In summary, night ventilation in summer in hollow core slab has two significant advantages. On the one hand, it can remove the heat released by PCM and improve the latent heat utilization rate. On the other hand, the cooling energy of the low-temperature outdoor air can be stored in the roof mass and directly transferred into the room to reduce the temperature of the roof's inner surface of at night. It is concluded that the most suitable ventilation speed is 2.4~2.5 m/s. In this case, the average temperature of the roof's inner surface can reach its lowest and the latent heat utilization rate of PCM can reach its highest.

## 5. Discussions

The effects of the PCM and ventilation have been investigated and analyzed. The amount of heat transfer from the roof to the room is significantly reduced during the daytime and the cooling energy of low-temperature air can be stored by ventilation at night by full use of the thermal storage performance of PCM and the concrete slab. The application of PCM significantly reduces the peak inner surface temperature of the roof and increases the time lag.

The thermal performance of VRFP is better than that of the conventional roof without PCM and ventilation. For VRFP, with the increase of the ventilation speed, the average utilization rate of latent heat and time lag are increased. When the airflow rate is 0.4, 0.8, 1.2, 2.0 and 2.4 m/s, the average latent heat utilization rate is 90.76%, 95.71%, 98.57%, 98.82% and 98.93%, the decrement factor is 0.035, 0.055, 0.061, 0.060 and 0.047 and the time lag is 9 h, 11 h, 11 h, 11 h, 12 h, respectively. Compared with nonventilation, when the ventilation speed is 0.4 m/s, the average utilization rate of latent heat of the PCM increases obviously; the decrement factor is the least which is 0.035. After the ventilation speed is increased to 1.2 m/s, the average utilization rate of latent heat of the PCM approaches 99%. When it is increased to 2.5 m/s, the average temperature decrease rate of the roof's inner surface is zero; it is the optimal ventilation speed. For conventional roof without PCM and ventilation, the average inner surface temperature of the roof is 30.98 °C, the peak temperature is 34.71 °C, the temperature amplitude is 7.01 °C, the time lag is 5 h and the decrement factor is 0.232. When the VRFP system adds 30 mm PCM layer with 35–37 °C melting temperature range, and is ventilated at 2.4 m/s, the average temperature of the roof's inner surface is 30.05 °C, the peak temperature is 30.60 °C, the temperature amplitude is decreased to 1.21 °C, the decrement coefficient is 0.047 and the time lag is 12 h. The thermal performance is improved obviously.

Compared with other types of roof, VRFP also has its superiority. Barrioset et al. [56] studied the thermal performance of five roofs. One of the structures is that the layers from exterior to interior are flexible elastomeric foam, high density concrete and lightweight plaster; the other one is an insulation roof and the layers are flexible elastomeric foam, expanded polystyrene, high density concrete and lightweight plaster, respectively. When solar absorption coefficient of out finish is 0.8, the decrement factors are 0.38 and 0.06, and the time lags are 5.2 h and 6.1 h, respectively. When the solar absorption coefficient of outer finish is 0.2, the decrement factors are 0.36 and 0.05 and the time lags are 5.3 h and 6.9 h, respectively. Normally, the roof with higher time lag and lower decrement factor shows better thermal performance. These results highlight that thermal performance of VRFP is much better than traditional roofs and there is also improvement compared with the insulation roof.

This structure is suitable for places with large temperature difference between day and night. In addition, there are some issues that need to be further studied, such as the optimal temperature difference between outdoor air and monitoring point to start ventilation, the simplified dynamic thermal network model and the annual energy saving potential prediction using the simplified model at each climatic region.

## 6. Conclusions

In this paper, a new structure named VRFP is proposed. The innovation point of this structure is that the cool air at night is used for ventilating through the duct embedded in the cast-in-place roof or the cavity of the precast concrete hollow-core slab in summer. The solidification heat can be effectively removed by the airflow, which leads to sufficient solidification of PCM at night. This structure can effectively reduce the influence of the outdoor environment on the indoor temperature, the inner surface temperature and air conditioning energy consumption. CFD numerical simulation was used. The effects of melting temperature, thickness of PCM layer and ventilation strategy on thermal performance of VRFP were studied under typical summer climate conditions in Wuhan. The conclusions are as follows:

(1) The PCM thickness has a great impact on the average temperature of inner surface, while the melting temperature has great impact on temperature amplitude. The PCM can obtain a reduction and a time-shift in the maximum heat flux through the roof.

(2) The PCM requires different thicknesses in terms of the melting temperature: the PCM with higher melting temperature needs a thinner thickness and the material with lower melting temperature needs a thicker thickness. Considering the effects of average temperature of inner surface, utilization rate of latent heat and decrement factor, the optimal melting temperature in Wuhan is 35~38 °C, and the suitable thickness of PCM layer is 30~40 mm.

(3) The ventilation starts from 22:00 to 23:00 in the summer of Wuhan and lasts for 6~8 h. With the increase of ventilation speed, the ventilation hours and the average inner surface temperature of the roof gradually decreases. The optimal ventilation speed range is 2.4~2.5 m/s, at this time the utilization rate of latent heat of PCM approaches 99%.

(4) Compared with the roof without PCM and ventilation, the VRFP with optimized design in Wuhan (30 mm thickness of PCM layer, 35~37 °C melting temperature and 2.4 m/s ventilation speed) has better thermal properties. The average temperature is reduced by 0.93 °C, which makes the energy consumption for cooling drop directly. The peak temperature is reduced by 4.11 °C, which promotes the efficiency of air conditioners by reducing the maximum load and improves indoor thermal comfort.

**Supplementary Materials:** The following are available online at http://www.mdpi.com/2071-1050/12/22/9315/s1, File, Additional work on CFD model establishment and grid independence analysis.

**Author Contributions:** Conceptualization, J.Y., and F.W.; methodology, J.Y.; software, K.L.; validation, J.Y., K.L. and H.Y.; formal analysis, H.Y.; investigation, F.W.; resources, J.Y., and F.W.; data curation, H.Y.; writing—original draft preparation, J.Y.; writing—review and editing, Y.L.; visualization, J.Y.; supervision, F.W.; project administration, J.Y., and F.W.; funding acquisition, F.W. All authors have read and agreed to the published version of the manuscript.

**Funding:** This work was funded by National Natural Science Foundation of China (grant number 51778255) and the Fundamental Research Funds for the Central Universities of China (grant number 2018KFYYXJJ131).

**Conflicts of Interest:** The authors declare no conflict of interest.

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
