# Peer review of "Simulation Study on Dynamic Thermal Performance of a New Ventilated Roof with Form-Stable PCM in Southern China"

_sustainability, doi:10.3390/su12229315_

Round 1
Reviewer 1 Report
This paper is well written and organized. However, for better understanding and generalization of main findings of this research, authors should address the followings in detailed manner.
- What kind of optimization approach has been applied for determining optimal melting temperature and thickness of the PCM.
- The result of this work looks like that the higher ventilation speed is the better for the system performance. What is the theoretical basis of the selected velocities you considered in this work.
- Authors need to provide more clear reasons for selecting the ranges of each parameter they considered in this work.
Reviewer 2 Report
Minor revision
The subject of the article is interesting, and the paper is well structured. The comments are as follow:
- In the literature review, other natural cooling methods like green roofs need to be mentioned.
- The paper's novelty is not clear. It is necessary to be added at the end of the introduction.
- Some figures about Ansys, such as mesh and results, need to be added in the appendix, not just graphs.
- It is better to add also the impact of PCM in the winter even by literature review analysis.
